# Substrate fluxes in brown adipocytes upon adrenergic stimulation and uncoupling protein 1 ablation

Sabine Schweizer[1], Josef Oeckl[1,2,3], Martin Klingenspor[1,2,3] (iD), Tobias Fromme[1,2] (iD)

**Brown adipocytes are highly specialized cells with the unique metabolic ability to dissipate chemical energy in the form of heat. We determined and inferred the flux of a number of key catabolic metabolites, their changes in response to adrenergic stimulation, and the dependency on the presence of the thermogenic uncoupling protein 1 and/or oxidative phosphorylation. This study provides reference values to approximate flux rates from a limited set of measured parameters in the future and thereby allows to evaluate the plausibility of claims about the capacity of metabolic adaptations or manipulations. From the resulting model, we delineate that in brown adipocytes (1) free fatty acids are a significant contributor to extracellular acidification, (2) glycogen is the dominant glycolytic substrate source in the acute response to an adrenergic stimulus, and (3) the futile cycling of free fatty acids between lipolysis and re-esterification into triglyceride provides a mechanism for uncoupling protein 1–independent, non-shivering thermogenesis in brown adipocytes.**

## Introduction

Brown adipocytes are highly specialized cells with the unique metabolic ability to dissipate chemical energy in the form of heat, thereby providing a means of non-shivering thermogenesis. This process is mediated by uncoupling protein 1 (UCP1) located in the inner mitochondrial membrane, where it uncouples respiration from adenosine triphosphate synthesis. Upon cold exposure, sympathetically released norepinephrine induces lipolysis. Released free fatty acids (FFAs) serve as both activators and fuel of non-shivering thermogenesis (for a review, see Cannon & Nedergaard [2004], Klingenspor et al [2017]). Adrenergic stimulation thus leads to strong changes in substrate fluxes that can be quantified to deduce thermogenic capacity. Oxygen consumption has long been established as a crucial metabolic parameter to delineate the components of coupled and uncoupled respiration in brown adipocytes. This type of measurement has become increasingly popular with

the availability of new technology including high-resolution respirometry and multi-plate–based assay systems. For cultured brown adipocytes, specific protocols have been developed to study mitochondrial oxygen consumption and quantify UCP1-mediated leak respiration (Li and Fromme et al, 2014, 2017). Based on previous knowledge on biochemical pathways, complementary parameters are often deduced from measured ones. For instance, extracellular acidification rate (ECAR) is often directly converted into glycolytic flux. As obvious as this procedure may seem, it is prone to be distorted by carbonic acid production by as much as 3%–100% of extracellular acidification, depending on cell type and substrate (Mookerjee et al, 2015). It is self-evident that even more caution is advisable when the estimate involves several assumptions, for example, a rate of triglyceride mobilization from glucose uptake (Virtanen et al, 2009). To aid future such calculations, we determined and inferred the flux of a number of key catabolic metabolites, their changes in response to adrenergic stimulation, and the dependency on the presence of UCP1 and/or oxidative phosphorylation. We demonstrate the strength of the resulting model by proof-of-principle conversions and with novel conclusions pertaining extracellular acidification, glycogen mobilization, and FFA cycling.

## Materials and Methods

### Animals and primary cell culture

Male 129Sv/S1 Ucp1-WT and KO mice, aged 5–6 wk and bred at the animal facility of Technical University of Munich in Weihenstephan, were used to prepare primary cultures of brown adipocytes. Brown fat precursor cells were isolated from interscapular brown adipose tissue as previously described (Li et al, 2014). After reaching confluency, induction medium (DMEM with 25 mM glucose (D5796; Sigma-Aldrich), 10% FBS, 0.5 mM isobutylmethylxanthine, 125 nM indomethacin, 1 mM dexamethasone, 850 nM insulin, 1 nM T3, and 1 µM rosiglitazone) was added for 2 d. Afterwards, cells were maintained in differentiation medium (DMEM with 25 mM glucose, 10% FBS, 850 nM insulin, 1 nM T3, 1 µM rosiglitazone) for 7 d. The medium was changed every 2 d. On day 7 of differentiation, the

[1]Chair of Molecular Nutritional Medicine, TUM School of Life Sciences, Technical University of Munich, Freising, Germany  [2]EKFZ—Else Kröner-Fresenius Center for Nutritional Medicine, Technical University of Munich, Freising, Germany  [3]ZIEL—Institute for Food and Health, Technical University of Munich, Freising, Germany

Correspondence: fromme@tum.de

medium was changed to a glucose concentration of 5.55 mM (supplemented to DMEM w/o glucose, 11966025; Gibco) at which all downstream experiments were performed. Conversion factors for protein, DNA, and cell number were established in 12-well plates. Protein was measured with Pierce BCA Protein Assay Kit (Thermo Fisher Scientific). DNA was detected fluorometrically (CyQUANT; Thermo Fisher Scientific). Cell number was deduced from DNA content assuming a DNA content of 6 pg per cell. Per $cm^2$, cultures contained an average 89 $\mu$g protein and 82,916 pg DNA = 13,819 cells.

### Respiration measurements

Oxygen consumption rate (OCR), ECAR, and proton production rate (PPR) were measured at 37°C using a XF-96 extracellular flux analyzer (Agilent Technologies). At day 7 of differentiation, the medium was replaced with prewarmed unbuffered DMEM (5.5 mM glucose) supplemented with 2, 3, or 4% essentially fatty acid–free BSA and incubated at 37°C in a non–carbon dioxide ($CO_2$) incubator for 1 h. Basal respiration was measured in untreated cells. Coupled respiration was inhibited by oligomycin treatment (5 $\mu$M). UCP1-mediated uncoupled respiration was determined after isoproterenol (0.5 $\mu$M) stimulation. Maximum respiratory capacity was assessed after carbonyl cyanide-p-trifluoromethoxyphenylhydrazone (Sigma-Aldrich) stimulation (7 $\mu$M). Finally, mitochondrial respiration was blocked by antimycin A (Sigma-Aldrich) (5 $\mu$M), leaving only non-mitochondrial OCR to be measured. OCR, ECAR, and PPR were automatically calculated by the Seahorse XF-96 software.

### Metabolite quantification

Lactate concentration in supernatants was determined using an enzymatic assay (Olsen, 1971). Perchloric acid, hydrazine sulphate, hydrazine hydrate, $NAD^+$, and lactate dehydrogenase were purchased from Sigma-Aldrich. Measurements were performed in a microplate reader (TECAN, infinite M200). Conditioned medium was analyzed for FFA and glycerol content by commercial kits (Wako Chemicals and Sigma-Aldrich, respectively). Measurements were performed using a microplate reader (TECAN, infinite M200) in accordance to manufacturer instructions. Intracellular glycogen concentration was determined by means of a commercial kit system (MET-5023; Cell Biolabs).

### Glucose uptake

1 d before the experiment insulin was removed from medium. After serum starvation (3 h), cells were stimulated with isoproterenol for 30 min. Then, the medium was changed to fresh assay medium (5.5 mM unlabeled glucose) containing stimulants and trace amounts (3 μCi) of 2-deoxy-d-[1-3H]-glucose. After three minutes, glucose uptake was terminated by washing cells with ice-cold stop solution (PBS, 25 mM glucose). Cells were lysed with 0.2 M NaOH at 60°C for 1 h. Incorporated radioactivity was determined by liquid scintillation counting (Perkin Elmer).

### Gene expression

Fully differentiated brown adipocytes were scraped into 200 μl Trisure (Bioline), RNA extracted following the manual and further purified by a column system (SV Total RNA Isolation System; Promega). We constructed libraries (TruSeq RNA LiBrary Preparation Kit v2; Illumina) and sequenced each sample to a depth of 10–20 million single reads 50 bp in length (HiSeq 2500; Illumina). We mapped reads to the mouse genome (National Center for Biotechnology Information build 38) and assigned uniquely identified reads (~¾ of total reads) to known transcripts to provide abundance values in reads per kilo bas per million mapped reads (RPKM) (Genomatix Mining Station; Genomatix). Pairwise comparison was performed with the DESeq2 1.4.5 algorithm (implemented in the Genome Analyzer platform; Genomatix; uses Wald test for *P*-value generation). Original data and analyses have been deposited at Gene Expression Omnibus (GSE119873).

Western blot detection of UCP1 was performed with a primary rabbit antibody raised against hamster UCP1 (Meyer et al, 2004). We detected pan-actin with a commercial mouse monoclonal antibody (MAB1501; Merck Millipore). Secondary antibodies were coupled to an infrared fluorophore (goat-anti-rabbit IRDye 800CW, 925-32210; donkey-anti-mouse 680RD, 925-68072; LI-COR) and visualized on an imaging system (Odyssey; LI-COR).

### Calculations

Carbon dioxide ($CO_2$) is metabolically produced during substrate combustion. In aqueous solution, generated $CO_2$ gets hydrated to carbonic acid ($H_2CO_3$), which subsequently dissociates to hydrogen carbonate ($HCO_3^-$) and $H^+$ at physiological pH. The fraction of hydrated $CO_2$ was calculated with the help of the overall pKa of 6.093 for $CO_2$ in aqueous solution at 37°C (Garrett & Grisham, 2009; Mookerjee et al, 2015). Acidification by $CO_2$ was calculated from the OCR assuming a respiratory quotient of 0.7 and a net production of $H^+$ per consumed $O_2$ of 0.65 during the complete oxidation of palmitate. Dissolved $CO_2$ is not completely hydrated at equilibrium. Net $H^+$ production was calculated by multiplying maximal produced $H^+$ per that consumed by the fraction of $CO_2$ hydrated as described in Mookerjee et al (2015). Proton production by lactic acid and FFA was calculated from lactate and $FFA^-$ concentration in the supernatant, because in aqueous solutions of physiological pH, these organic acids quantitatively dissociate into their conjugated base and stoichiometrically release $H^+$. Maximal FFA amount used for $\beta$-oxidation was deduced from $O_2$ consumption. Calculations were based on the caloric equivalent (431 $\mu$J/nmol $O_2$) and Gibbs energy (9.8 μJ/pmol) of palmitate (Gnaiger & Kemp, 1990; Olmsted & Williams, 1997). The cycling rate/re-esterification rate was calculated as "(3 × glycerol release) – (FFA release + FFA oxidation)" as described previously (Brooks et al, 1982; Newsholme et al, 1983). The amount of hydrolyzed triglyceride was estimated from the total amount of FFAs ($FFA_{exported}$ + $FFA_{oxidized}$ + $FFA_{re-esterified}$)/3.

## Results

This study utilized a model of cultured, primary brown adipocytes from WT and Ucp1-KO mice. This cellular model was initially validated by RNA sequencing. Differentiated, brown adipocytes expressed the Ucp1 gene in a striking abundance. Of nearly 30,000

identified, RefSeq annotated transcripts, Ucp1 was the 17[th] most abundant. Accordingly, UCP1 protein could be robustly detected by Western blot (Fig S1A). A list of other typical brown adipocyte marker transcript were detected in appreciable amounts (Fig S1). In terms of transcriptomics, brown adipocytes of WT and Ucp1-KO origin were virtually undistinguishable. Of nearly 30,000 analyzed genes, 20 were differentially expressed, among them Ucp1 (Table S1). The remaining mRNA in the Ucp1-KO cells was attributable to a non-functional remnant transcript lacking exons 2 and 3 (Fig S1B). None of the 19 other differential genes were recognized to be functionally associated with significant metabolic pathways. Consequently, we detected no differences in transcripts indicative of brown adipocyte identity and/or mitochondrial mass or function (Fig S1C and D). Thus, primary brown adipocytes as employed in this study constitute a suitable model to study brown adipocyte function and to specifically detect the consequences of Ucp1 ablation. In this model, we measured and deduced key metabolic parameters in the basal state and upon stimulation with the $\beta$-adrenergic agonist isoproterenol. From these, we created an interconnected, quantitative model of major catabolic pathways.

The accepted model of adrenergically activated non-shivering thermogenesis comprises intracellular signaling cascades converging on lipolytic enzymes and regulators to increase cytosolic, and consequently mitochondrial, FFA levels. These FFAs serve a double role as both oxidative fuel and steric activators of the unique, thermogenic UCP1 (for a review, see Klingenspor et al [2017]). Indeed, adrenergic activation led to a strong increase in lipolysis. The deduced rate of triglyceride mobilization increased more than fourfold to 28.2 nmol × h$^{-1}$ × cm$^{-2}$ (= 22.9 $\mu$g × h$^{-1}$ × cm$^{-2}$ = 1.7 ng × h$^{-1}$ × cell$^{-1}$ = 257.9 ng × h$^{-1}$ * $\mu$g protein$^{-1}$; assuming tripalmitin, see the Materials and Methods section for conversion factors). We found the vast majority of liberated FFAs released into the medium (Fig 1 and Table 1). Based on oxygen consumption, less than 10% of FFAs were subject to mitochondrial $\beta$-oxidation even when assuming exclusive lipid oxidation. This observation is in line with a high amount of glycerol released by activated adipocytes.

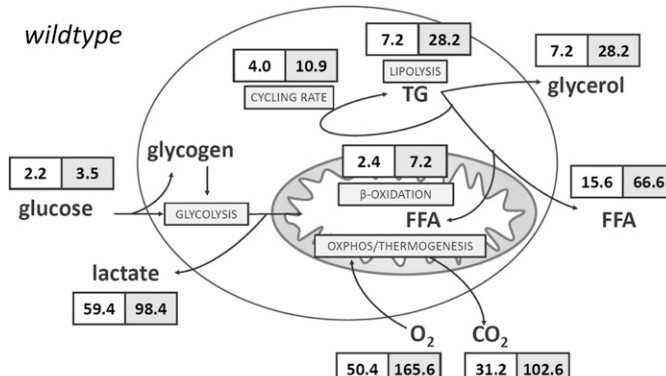

**Figure 1. Substrate fluxes in WT brown adipocytes.**
Substrate fluxes for major catabolic pathways in WT murine brown adipocytes. A white background indicates basal flux rate and a grey background indicates isoproterenol-stimulated rates. All data are provided in nmol × h$^{-1}$ × cm$^{-2}$, see the Materials and Methods section for calculations and conversion factors to the per cell and per protein level, and Table 1 for complete dataset. TG, triglyceride mobilized.

**Table 1. Complete dataset of measured and calculated parameters based on three independent experiments with primary brown adipocytes from WT and Ucp1-KO mice.**

| | Ucp1-KO | WT |
|---|---|---|
| **Basal** | | |
| Glycerol release | 6.0 ± 1.8 | 7.2 ± 1.8 |
| FFA release | 9.6 ± 1.8 | 15.6 ± 7.2 |
| FFA oxidized | 1.8 ± 0.0 | 2.4 ± 0.6 |
| FFA re-esterified | 6.4 ± 4.6 | 4.0 ± 1.8 |
| TAG hydrolyzed | 5.9 ± 1.7 | 7.2 ± 2.3 |
| O$_2$ consumption | 44.4 ± 6.6 | 50.4 ± 19.2 |
| CO$_2$ production | 27.6 ± 4.2 | 31.2 ± 12.0 |
| Glucose uptake | 2.3 ± 0.0 | 2.2 ± 0.0 |
| Lactate release | 114.6 ± 15.0 | 59.4 ± 7.2 |
| **Iso** | | |
| Glycerol release | 58.2 ± 9.0 | 28.2 ± 5.4 |
| FFA release | 124.8 ± 24.0 | 66.6 ± 3.6 |
| FFA oxidized | 6.6 ± 0.0 | 7.2 ± 0.6 |
| FFA re-esterified | 43.7 ± 4.6 | 10.9 ± 19.3 |
| TAG hydrolyzed | 58.4 ± 8.7 | 28.2 ± 5.1 |
| O$_2$ consumption | 151.8 ± 6.6 | 165.6 ± 10.8 |
| CO$_2$ production | 94.2 ± 4.2 | 102.6 ± 6.6 |
| Glucose uptake | 3.8 ± 0.6 | 3.5 ± 0.6 |
| Lactate release | 165.6 ± 24.6 | 98.4 ± 14.4 |
| **Oligo basal** | | |
| Glycerol release | 3.0 ± 2.4 | 4.2 ± 1.2 |
| FFA release | 8.4 ± 1.8 | 11.4 ± 6.0 |
| FFA oxidized | 0.6 ± 0.0 | 0.6 ± 0.0 |
| FFA re-esterified | <0 | 0.6 ± 3.0 |
| TAG hydrolyzed | 3.0 ± 2.2 | 4.4 ± 1.0 |
| O$_2$ consumption | 20.4 ± 4.8 | 18.6 ± 6.6 |
| CO$_2$ production | 12.6 ± 3.0 | 11.4 ± 4.2 |
| Lactate release | 156.6 ± 19.2 | 102.6 ± 25.2 |
| **Oligo iso** | | |
| Glycerol release | 38.4 ± 6.6 | 16.2 ± 3.6 |
| FFA release | 123.6 ± 21.6 | 38.4 ± 7.2 |
| FFA oxidized | 2.4 ± 0.0 | 6.6 ± 1.8 |
| FFA re-esterified | <0 | 3.9 ± 4.2 |
| TAG hydrolyzed | 38.2 ± 6.6 | 16.3 ± 4.0 |
| O$_2$ consumption | 48.0 ± 2.4 | 152.4 ± 46.8 |
| CO$_2$ production | 30.0 ± 1.2 | 94.8 ± 29.4 |
| Lactate release | 181.8 ± 16.2 | 126.6 ± 3.6 |

The mean values ± SD (nmol/h/cm$^2$) in the basal state, stimulated with isoproterenol (iso) and/or in the presence of the ATP synthase inhibitor oligomycin (oligo), are shown. Glycerol, FFAs, and lactate levels were measured in the supernatant medium. Oxygen consumption was measured in an XF96 extracellular flux analyzer. CO$_2$, oxidized FFA, and hydrolyzed triglycerides (TAG) were calculated from respiration, based on an assumed respiratory quotient of 0.7.

The ratio of glycerol to FFAs detected in the medium is lower than the stoichiometrically expected value of three also when taking into account oxidized FFAs. Thus, 5–20% of FFAs remained unaccounted for and represented FFAs re-esterified into triglyceride (Brooks et al, 1982; Newsholme et al, 1983).

Adrenergically increased lipolysis activated non-shivering thermogenesis in line with the accepted model (see above). Oxygen consumption and deduced carbon dioxide ($CO_2$) production increased more than threefold (Fig 1 and Table 1). Notably, $CO_2$ can react with water molecules to form carbonic acid, which in turn may dissociate forming carbonate and hydronium ions. By this route, respiration is a significant contributor to overall PPR as measured in our assay (Mookerjee et al, 2015). We compared total PPR with the release rate of possible source molecules. Carbonic acid, lactic acid, and FFAs were sufficient to explain total PPR plausibly, demonstrating the validity of our model (Fig 2). For the first time, we demonstrate that FFAs are a significant contributor to PPR of brown adipocytes. While under basal conditions lactic acid is indeed a major source of protons, it does not play a dominant role in the adrenergically induced doubling in proton production, which is similarly caused by both FFA and carbonic acid release. To explore this concept further, we artificially increased glycolytic flux by inhibiting mitochondrial ATP synthase with oligomycin. Indeed, this manipulation increased the contribution of glycolytic lactic acid to PPR in the basal state (Fig 2). The adrenergic increase in proton production, however, remained being caused by FFA and carbonic acid release. Taken together, all three proton sources must be taken into account when studying proton production of adipocytes to avoid misinterpretations.

The numerical accordance of release rates from proton source molecules with observed total proton production clearly validates our quantification of the contributing metabolites. Lactate release, however, was markedly higher than glucose uptake in the basal and in the stimulated state (Fig 1 and Table 1). Because lactic acid is a product of glycolysis, this surprising observation requires an additional source of glucose that escaped quantification in our uptake assay. The only plausible candidate seemed to be intracellular glycogen, which has been reported to be present in unusually high amounts in brown as compared with white adipocytes and may even exceed the glycogen content of muscles and

the liver (Creasey & Gray, 1951; Farkas et al, 1999; Carmean et al, 2013; Carmean, 2015). We determined glycogen content in brown adipocytes deprived of glucose to monitor changes caused by β-adrenergic activation (Fig 3A). Indeed, we observed a very high glycogen content, as high as 1 ng per cell (for comparison: a 50-μm diameter sphere of water weighs 65.5 ng), that strongly decreased after isoproterenol treatment (Fig 3A). Translated into glucose units, this change constituted a flux of −54 nmol × h$^{-1}$ × cm$^{-2}$ and thus plausibly accounted for the bulk of excess lactate not explained by glucose uptake (Fig 1). Interestingly, in the presence of high glucose concentrations, an initially lower glycogen reserve is not depleted, but rather increased by isoproterenol (Fig 3B). In the absence of proportionally increased glucose uptake, this seems to imply glycogen formation from amino acid–derived gluconeogenesis, a phenomenon plausible in regard to amino acid abundance in DMEM cell culture medium, but of no obvious purpose in this context. It remains to be further explored in the future.

Adaptive, non-shivering thermogenesis in brown adipocytes depends on the presence of UCP1. In the absence of this thermogenic protein, adrenergic signaling still leads to increased lipolysis but fails to stimulate respiration (Matthias et al, 2000; Li et al, 2014), in line with the role of FFAs as steric UCP1 activators. In other words, lipolytic activity is positioned upstream of respiration on the adrenergic signaling axis. We applied this rationale extending it to all parameters determined in our model by repeating the experiment with brown adipocytes isolated from Ucp1-KO mice. To our surprise, every single parameter assessed in our model reacted to isoproterenol treatment in a similar qualitative pattern as WT cells (Fig 3 and Table 1). In the case of lipolytic rate and glucose uptake, this independence of UCP1 was not entirely unexpected and has been discussed before (Li & Fromme et al, 2014; Hankir et al, 2017). As far as respiration is concerned, our observation seemed in diametrical opposition to earlier findings (Matthias et al, 2000; Li & Fromme et al, 2014, 2017). We excluded a FFA-caused uncoupling artefact by increasing the amount of FFA-buffering bovine serum albumin to excessive amounts without any reduction in respiration (Fig S2). The only difference between the model presented here and our own contradicting previous assessment (Li & Fromme et al, 2014) is the presence/absence of the ATP synthase inhibitor oligomycin. Indeed, repeating our measurements in the presence of oligomycin abrogated all UCP1-independent respiration in response to an adrenergic stimulus (Table 1). It follows that isoproterenol treatment led to a drastic increase in ATP turnover in the primary, Ucp1-KO brown adipocyte model leading to a respiration rate similar to UCP1-mediated uncoupling. We corroborated this conclusion by a dedicated experiment in which we treated fully differentiated, brown adipocytes of WT or Ucp1-KO origin either with oligomycin and isoproterenol or in a reversed order. In line with previous reports (Keipert & Jastroch, 2014; Keipert et al, 2017), isoproterenol-induced respiration was fully sensitive to oligomycin in Ucp1-KO cells, but only partially reduced in WT cells (Fig 5A). This mechanism effectively provided a means of non-shivering thermogenesis independent from UCP1.

To identify the nature of the adrenergically induced ATP sink, we re-analyzed our dataset for a pattern of (1) isoproterenol-induced and (2) oligomycin-sensitive changes that are (3) specific for Ucp1-KO cells. The only parameter fitting this description is the

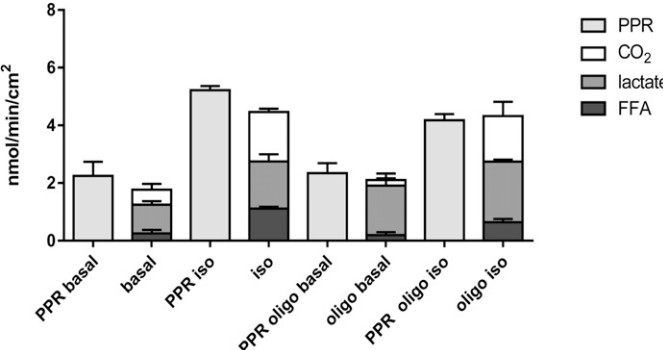

**Figure 2. Contributors to proton production.**
PPR as measured in the basal and the isoproterenol (iso)-induced state (uniform light grey bars) and as calculated from three different source molecules (stacked bars). Oligo–oligmycin. All data are mean values ± SD, n = 3.

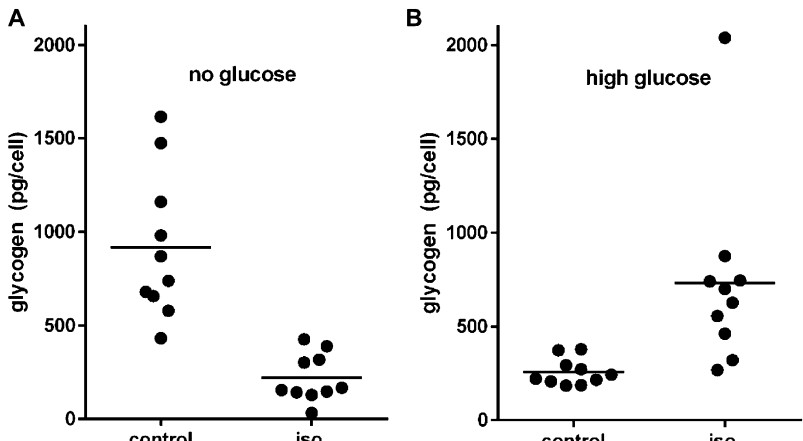

**Figure 3. Glycogen content of brown adipocytes.**
Altered glycogen content of cultured brown adipocytes in response to isoproterenol (iso) treatment **(A)** in the absence of glucose and **(B)** in a medium containing 25 mM glucose.

re-esterification rate (Fig 4 and Table 1), that is, the fraction of lipolytically liberated FFAs that is subsequently re-esterified into triglyceride instead of being released or oxidized. In general, lipolysis is stimulated by isoproterenol in both WT and Ucp1-KO cells and both in the absence and presence of oligomycin (Table 1). In WT brown adipocytes, the ratio of FFAs to glycerol released is in the range of 2.44–2.85 in all conditions tested. In Ucp1-KO cells, however, FFA to glycerol ratios are much lower (1.92–2.25), but increase drastically in the presence of oligomycin (2.79–3.54). Obviously, FFAs are ATP-dependently re-esterified to glycolytically derived G3P (Fig 5B). This model is supported by Ucp1-KO cells releasing more lactate in the presence of oligomycin than WT cells, in line with an increased glycolytic rate to provide G3P (Table 1).

Taken together, we describe a model of key catabolic pathways in WT and Ucp1-KO brown adipocytes at baseline and stimulated by isoproterenol. Comparison of substrate fluxes with ECAR revealed FFAs as a significant proton source. We further identified high intracellular glycogen stores as a preferred initial source for glycolytic substrate under adrenergic stimulation. Finally, respiration of Ucp1-KO brown adipocytes was stimulated β-adrenergically to similar levels as in WT brown adipocytes. This UCP1-independent thermogenic component requires ATP and likely involves the re-esterification of FFAs to glycolytically provided G3P. A futile cycle of lipolysis and simultaneous re-esterification may provide the ATP sink required for the observed thermogenic effect.

## Discussion

We present an interconnected, quantitative model of major catabolic pathways in cultured, primary brown adipocytes. This model of metabolic fluxes describes and predicts how an adrenergic stimulus changes key metabolic reactions in brown fat cells. Because all parameters were determined in the presence and absence of the thermogenic protein UCP1, our model differentiates both β-adrenergic and UCP1-dependent thermogenic effects. Primary data generated (Table 1 and transcriptomic dataset) can serve as a reference to quantitatively evaluate the plausibility of claims about the capacity of metabolic adaptations or manipulations.

We employed the model to clarify the predictive value of extracellular acidification (ECAR) with regard to glycolytic flux. Glycolytically produced lactic acid is often considered the major source of medium acidification and ECAR is commonly used as a direct and quantitative measure of glycolytic flux (Wu et al, 2007; Nadanaciva et al, 2012; Mookerjee et al, 2015; Ramirez et al, 2017). As others have pointed out before, carbon dioxide–derived carbonic acid is a second important contributor to extracellular acidification complicating this direct conversion (Mookerjee et al, 2015). In a cellular model of adipocytes, FFAs represent a third major acidification source. Proton production under adrenergic stimulation is even predominantly caused by metabolically produced FFAs and $CO_2$, and not by lactic acid. In cultured adipocytes and/or in response to lipolytic stimuli, ECAR is thus not a suitable measure of glycolytic flux.

We noticed a stark discrepancy between glucose uptake rate and glycolytic lactate production. The latter seemed to require a substantial, additional glucose source that we confirmed to be intracellular glycogen. An exceptionally high glycogen content of brown adipocytes reaching levels comparable with fed liver has

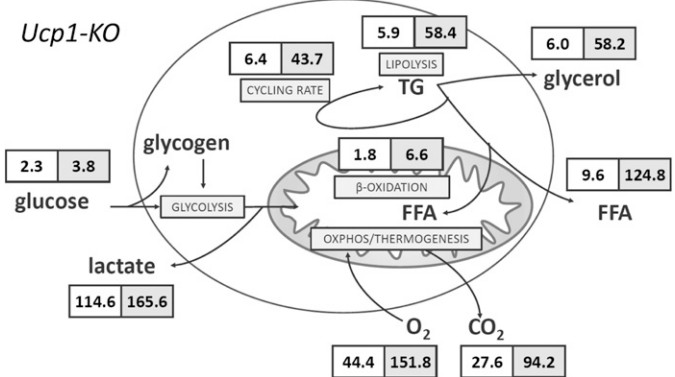

**Figure 4. Substrate fluxes in Ucp1-KO brown adipocytes.**
Substrate fluxes for major catabolic pathways in Ucp1-KO murine brown adipocytes. A white background indicates basal flux rate and a grey background indicates isoproterenol-stimulated rates. All data are provided in nmol × h⁻¹ × cm⁻², see the Materials and Methods section for calculations and conversion factors to the per cell and per protein level and Table 1 for complete dataset. TG, triglyceride mobilized.

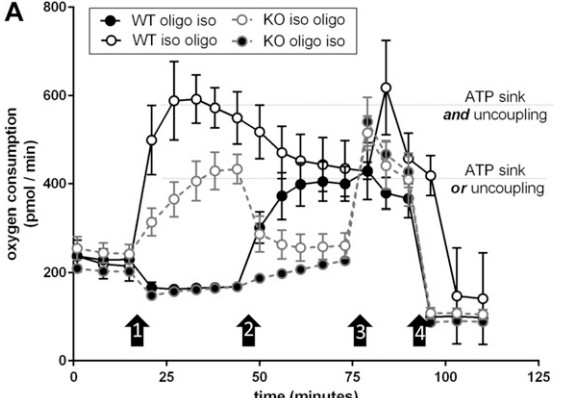

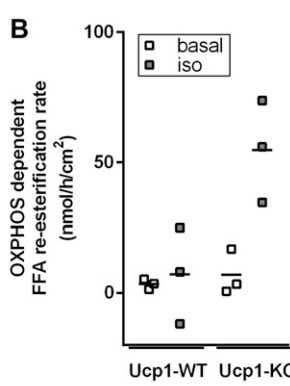

**Figure 5. Thermogenic respiration in WT and Ucp1-KO brown adipocytes.**
**(A)** Oxygen consumption of cultured brown adipocytes isolated from WT or Ucp1-KO mice. Black arrows indicate injections. Injections 1 and 2 were either oligomycin (oligo) or isoproterenol (iso) (see legend), injection 3 was carbonyl cyanide-p-trifluoromethoxyphenylhydrazone (FCCP), and injection 4 was antimycin A. Shown are mean values ± SD, n = 4–11. **(B)** FFA re-esterification rate depending on oxidative phosphorylation in cultured brown adipocytes isolated from WT or Ucp1-KO mice in the basal or the isoproterenol (iso)-induced state.

been reported before (Creasey & Gray, 1951; Farkas et al, 1999; Carmean et al, 2013; Carmean, 2015). It seems that at least in culture, brown adipocytes obtain a large portion of glycolytic substrate from intracellular stores at the expense of imported glucose. This phenomenon has to be taken into account when quantifying glucose utilization by glucose uptake measurements. Significantly, the expanse and intensity of 18-fluorodeoxyglucose uptake as a marker of active brown fat determined by positron emission tomography may depend on glycogen stores and thus on the fasting state and blood glucose levels.

In brown adipocytes, adrenergic signaling leads to increased lipolysis and activation of UCP1. Most parameters that were assessed in our model such as glucose uptake, triglyceride degradation, or FFA and glycerol release were elevated in response to isoproterenol treatment, irrespective of the presence or absence of UCP1. Interestingly, this was also true for oxygen consumption. A $\beta$-adrenergic agonist caused a dramatic increase in ATP turnover in Ucp1-KO brown adipocytes, leading to an oligomycin sensitive respiration undistinguishable from UCP1-mediated uncoupling. Importantly, although Ucp1-KO mice are sensitive to acute cold exposure (Enerbäck et al, 1997), they are able to recruit UCP1-independent thermogenic mechanisms and defend their body temperature when gradually adapting to cold, exhibiting almost 50% of the Ucp1-WT increase in maximal cold-induced heat production (Golozoubova et al, 2001; Ukropec et al, 2006a, b; Meyer et al, 2010; Shabalina et al, 2010). Multiple mechanisms of adaptive heat production have been proposed to provide this compensatory capacity, including creatine-dependent substrate cycling in brite adipocytes (Bertholet et al, 2017; Kazak et al, 2015), futile cycling of calcium between ER and cytosolic compartments in white adipose tissue (Ukropec et al, 2006a, b) or skeletal muscle (Bal et al, 2012; Rowland et al, 2015) and, importantly, futile substrate cycling in white adipocytes based on parallel lipolysis and re-esterification (Granneman et al, 2003; Flachs et al, 2013; Mottillo et al, 2014; Flachs et al, 2017). Our own data clearly corroborate the latter model and involve the circular processes lipolysis, liberation of glycerol and FFAs, and re-esterification of FFAs to glycolytically produced glycerol, effectively providing an ATP sink that meets the requirements to be stimulated adrenergically, oligomycin-sensitive and independent of UCP1. It remains to be demonstrated whether the process we and others

observe in primary, cultured brown adipocytes is in fact present and physiologically relevant in vivo. At least in directly isolated mature brown adipocytes, adrenergic stimulation does not seem to lead to increased respiration (Matthias et al, 2000). On the other hand, adipocyte triglyceride synthesis is required for cold-induced thermogenesis in vivo, whereas UCP1 is not (Ellis et al, 2010; Meyer et al, 2010; Keipert et al, 2017). A key experiment will be to determine whether brown adipose tissue, white adipose tissue, muscle, or any other organ system is the actual source of adaptive, non-shivering thermogenic capacity in Ucp1-KO mice. As far as brown fat is concerned, optoacoustic imaging of hemoglobin gradients may provide a suitable methodology to resolve this question in the future (Reber et al, 2018).

Intriguingly, gene expression is virtually undistinguishable in WT and Ucp1-KO cells. All components of powerful, UCP1-independent thermogenesis by futile lipolysis and re-esterification are obviously already present in sufficient amounts in brown adipocytes. On the one hand, this renders the detection of said mechanism by transcriptomic methods difficult or impossible. On the other hand, it suggests that no additional component is required and could thereby be invoked in any cell of sufficient lipolytic/re-esterification capacity, possibly even non-adipocytes.

In summary, this study provides quantitative, interconnected substrate fluxes in brown adipocytes and resolves their dependency on adrenergic stimulation, the presence of UCP1 and the ability to phosphorylate ATP. It provides reference values to approximate flux rates from a limited set of measured parameters in the future and thereby allows to evaluate the plausibility of claims about the capacity of metabolic adaptations or manipulations. We furthermore delineate from our model that in brown adipocytes (1) FFAs are a significant contributor to extracellular acidification, (2) glycogen is the dominant glycolytic substrate source in the acute response to an adrenergic stimulus, and (3) the futile cycling of FFAs between lipolysis and re-esterification into triglyceride provides a mechanism for UCP1-independent, non-shivering thermogenesis in brown adipocytes.

## Supplementary Information

# Acknowledgements

This work was supported by a grant by the Deutsches Zentrum für Diabetesforschung (01GI0923). Publication was supported by the German Research Foundation and the Technical University of Munich within the funding program Open Access Publishing. We thank Sabine Mocek for excellent technical assistance.

## Author Contributions

S Schweizer: data curation, formal analysis, validation, investigation, visualization, methodology, and writing—review and editing.
J Oeckl: formal analysis, investigation, visualization, methodology, and writing—review and editing.
M Klingenspor: conceptualization, resources, supervision, funding acquisition, project administration, and writing—review and editing.
T Fromme: conceptualization, data curation, software, formal analysis, supervision, validation, visualization, and writing—original draft, review, and editing.

## Conflict of Interest Statement

The authors declare no conflicting interests.

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
