## [Reviewer comments · Life Science Alliance]

Life Science Alliance

Substrate fluxes in brown adipocytes upon adrenergic stimulation and uncoupling protein 1 ablation

Sabine Schweizer, Josef Oeckl, Martin Klingenspor, and Tobias Fromme
DOI: 10.26508/lsa.201800136

Corresponding author(s): Tobias Fromme, Technische Universität

Review Timeline:	Submission Date:	2018-07-20
	Editorial Decision:	2018-08-15
	Revision Received:	2018-10-22
	Editorial Decision:	2018-11-02
	Revision Received:	2018-11-05
	Accepted:	2018-11-05

Scientific Editor: Andrea Leibfried

Transaction Report:

August 15, 2018

Re: Life Science Alliance manuscript #LSA-2018-00136-T

Dr. Tobias Fromme
Technische Universität
Molecular Nutritional Medicine
Gregor Mendel Str. 2
Freising-Weihenstephan 85350
Germany

Dear Dr. Fromme,

Thank you for submitting your manuscript entitled "Substrate fluxes in murine brown adipocytes and their dependency on adrenergic stimulation, uncoupling protein 1 and oxidative phosphorylation" to Life Science Alliance. The manuscript was assessed by expert reviewers, whose comments are appended to this letter. We invite you to submit a revision if you can address the reviewers' key concerns, as outlined here.

As you will see, the reviewers think that your work could be of value to others in the field. However, they both request additional controls to make sure that your cell culture-based results are not biased by differential expression or differentiation levels, or by an altered signaling response to adrenergic stimulation. Reviewer #2 furthermore notes that the glucose concentration used is non-physiological, and we encourage you to re-assess the noted effects with a lower concentration as suggested by the reviewer.

- A letter addressing the reviewers' comments point by point.
- An editable version of the final text (.DOC or .DOCX) is needed for copyediting (no PDFs).
- High-resolution figure, supplementary figure and video files uploaded as individual files: See our detailed guidelines for preparing your production-ready images, <http://life-science-alliance.org/authorguide>
- Summary blurb (enter in submission system): A short text summarizing in a single sentence the

study (max. 200 characters including spaces). This text is used in conjunction with the titles of papers, hence should be informative and complementary to the title and running title. It should describe the context and significance of the findings for a general readership; it should be written in the present tense and refer to the work in the third person. Author names should not be mentioned.

B. MANUSCRIPT ORGANIZATION AND FORMATTING:

Full guidelines are available on our Instructions for Authors page, <http://life-science-alliance.org/authorguide>

Thank you for this interesting contribution to Life Science Alliance. We are looking forward to receiving your revised manuscript.

Sincerely,

Reviewer #1 (Comments to the Authors (Required)):

This is an interesting and timely paper which provides a comprehensive evaluation of metabolic

pathways acting in mature brown fat cells. The study measures metabolic activity in both wildtype and UCP1 KO adipocytes in the presence or absence of β -adrenergic stimulation. In addition to providing a valuable resource for the field, the study provides two significant findings: (1) glycogen is a key source for glycolysis during acute activation; (2) UCP1 KO cells engage high levels of futile cycling between FA lipolysis and re-esterification; this appears to account for high levels of ATP-dependent respiration/thermogenesis. I find the paper to be very compelling and timely, given the recent focus on UCP1-independent pathways. A role for futile cycling of FA between lipolysis and re-esterification has been reported/discussed, but has received relatively little attention. The quantitative assessments in this paper provide strong evidence that this pathway is likely to play a significant role. The only minor suggestion would be to provide some additional detail on the cell models. How well differentiated are the adipocytes and is UCP1 well-expressed. Do the WT and UCP1 KO cells have equivalent levels of mitochondria, etc. Overall, I think this is an important contribution to the field that should have significant impact. I recommend that this be published without delay.

Reviewer #2 (Comments to the Authors (Required)):

Schwizer and colleagues have explored different metabolic fluxes in differentiated primary adipocytes in order to better understand the response to adrenergic stimulation and the role of UCP1 uncoupling. Their analyses include the cellular uptake and fate of glucose, as well as that of glycerol and FFA production upon lipolysis. Several interesting observations arise from their work. First, that adipocytes contain very significant amounts of glycogen that largely feed glycolysis during adrenergic stimulation. Second that extracellular acidification rates, often used as a proxy for anaerobic glycolysis, are not simply determined by lactate release in cultured adipocytes, but also by CO₂ and free fatty acids. Interestingly, the authors observe that UCP1 ablation leads to a larger rate of fatty acid re-esterification upon adrenergic stimulus, which could constitute an alternative way to enhance non-shivering thermogenesis in the absence of UCP1.

As the authors state, this work could be a valuable resource for metabolic flux analyses and can help preventing some long-standing misconceptions in the field (i.e.: what does ECAR really reflect). Nevertheless, it finds some pitfalls that deserve some attention:

1/ One of my major criticisms is that these analyses are performed in 25 mM glucose, which is a totally unphysiological setting. In turn, this might drive alterations in glycogen content and in lipid utilization, artificially prompting some of the most relevant observations. The authors could try to assess some of these parameters in 5 mM glucose concentrations or validate their observations to some degree using native BAT.

2/ The authors speculate that amino acid metabolism could drive the increased glycogen synthesis. Was the culture medium supplemented with (or rich in) glutamine?

3/ Several parameters were measured using different plating techniques. This is especially worrisome for Seahorse measurements. Many cell lines fail to properly differentiate in Seahorse plates. Could the authors provide some proof that cells differentiated in regular plate wells are comparable to adipocytes differentiated in Seahorse plates?

4/ The increased lipolysis in the UCP1KO adipocytes after adrenergic stimulation is spectacular, and twice as high as in wild-type adipocytes. The authors should verify that the signaling response to adrenergic stimuli (e.g: HSL or perilipin phosphorylation) is comparable between genotypes.

5/ The authors derive FAO from O₂ consumption. Why did they rule out that aerobic glycolysis could significantly contribute to O₂ consumption? Most reports to day indicate that BAT is an avid glucose consuming tissue. The contribution of FAO to global O₂ consumption could be estimated using etomoxir, albeit not being a perfect tool.

Reviewer #1 (Comments to the Authors (Required)):

This is an interesting and timely paper which provides a comprehensive evaluation of metabolic pathways acting in mature brown fat cells. The study measures metabolic activity in both wildtype and UCP1 KO adipocytes in the presence or absence of β -adrenergic stimulation. In addition to providing a valuable resource for the field, the study provides two significant findings: (1) glycogen is a key source for glycolysis during acute activation; (2) UCP1 KO cells engage high levels of futile cycling between FA lipolysis and reesterification; this appears to account for high levels of ATP-dependent respiration/thermogenesis.

I find the paper to be very compelling and timely, given the recent focus on UCP1-independent pathways. A role for futile cycling of FA between lipolysis and re-esterification has been reported/discussed, but has received relatively little attention. The quantitative assessments in this paper provide strong evidence that this pathway is likely to play a significant role.

The only minor suggestion would be to provide some additional detail on the cell models. How well differentiated are the adipocytes and is UCP1 well-expressed. Do the WT and UCP1 KO cells have equivalent levels of mitochondria, etc. Overall, I think this is an important contribution to the field that should have significant impact. I recommend that this be published without delay.

Thank you for the positive feedback. We have meanwhile better characterized gene expression of the cell model by transcriptomics and Ucp1 Western Blot. The manuscript has been modified accordingly. In essence, primary brown adipocytes abundantly express Ucp1 and other brown fat markers and there is no indication of significant changes in Ucp1-KO cells. These data has been included as a supplemental dataset. Original data can be downloaded at GEO (GSE119873).

Reviewer #2 (Comments to the Authors (Required)):

Schwizer and colleagues have explored different metabolic fluxes in differentiated primary adipocytes in order to better understand the response to adrenergic stimulation and the role of UCP1 uncoupling. Their analyses include the cellular uptake and fate of glucose, as well as that of glycerol and FFA production upon lipolysis. Several interesting observations arise from their work. First, that adipocytes contain very significant amounts of glycogen that largely feed glycolysis during adrenergic stimulation. Second that extracellular acidification rates, often used as a proxy for anaerobic glycolysis, are not simply determined by lactate release in cultured adipocytes, but also by CO₂ and free fatty acids. Interestingly, the authors observe that UCP1 ablation leads to a larger rate of fatty acid reesterification upon adrenergic stimulus, which could constitute an alternative way to enhance non-shivering thermogenesis in the absence of UCP1.

As the authors state, this work could be a valuable resource for metabolic flux analyses and can help preventing some long-standing misconceptions in the field (i.e.: what does ECAR really reflect). Nevertheless, it finds some pitfalls that deserve some attention:

1/ One of my major criticisms is that these analyses are performed in 25 mM glucose, which is a totally unphysiological setting. In turn, this might drive alterations in glycogen content and in lipid utilization, artificially prompting some of the most relevant observations. The authors could try to asses some of

these parameters in 5 mM glucose concentrations or validate their observations to some degree using native BAT.

We thank the reviewer to point out this rather embarrassing oversight and apologize. The 25mM glucose stated in the Materials and methods part refer to the maintenance medium of our primary cells (DMEM, Sigma-Aldrich D5796). All assays shown were uniformly performed at a glucose concentration of 5.55mM. We amended the MS to clearly state this fact.

2/ The authors speculate that amino acid metabolism could drive the increased glycogen synthesis. Was the culture medium supplemented with (or rich in) glutamine?

The assay medium was DMEM without glucose (Gibco 11966025) supplemented with 5.55mM glucose. This DMEM type contains regular glutamine (i.e. not the stabilized “glutamax” variety) at a concentration of 4mM, which is the routine glutamine concentration in DMEMs. Nevertheless, since assays were started with fresh media and DMEM is comparatively rich in amino acids (e.g. four-fold higher than the original basal medium Eagle), amino acids constitute a plausible, abundant substrate for gluconeogenesis (but of elusive purpose, as we point out in the MS). We added this piece of information into the respective discussion part.

3/ Several parameters were measured using different plating techniques. This is especially worrisome for Seahorse measurements. Many cell lines fail to properly differentiate in Seahorse plates. Could the authors provide some proof that cells differentiated in regular plate wells are comparable to adipocytes differentiated in Seahorse plates?

Adipogenic differentiation is indeed a challenge on Seahorse plates. Luckily, murine primary brown adipocytes are a pleasant exception to this rule and perform rather fine under all kinds of growth conditions. We here add an image for the reviewer, on which multilocular cells can be clearly recognized (see below; the shade in the upper left corner is one of the Seahorse plate spacer bumps). Apart from this visual evidence, we would like to point out that our data itself is good proof for widespread differentiation comparable under all conditions. Firstly, proton source molecules measured in different plate formats beautifully add up to total proton production rate (measured on Seahorse plates) and secondly, we show strong UCP1-specific, adrenergically induced respiration, which can only be explained by the presence of fully differentiated brown adipocytes.

4/ The increased lipolysis in the UCP1KO adipocytes after adrenergic stimulation is spectacular, and twice as high as in wild-type adipocytes. The authors should verify that the signaling response to adrenergic stimuli (e.g: HSL or perilipin phosphorylation) is comparable between genotypes.

We indeed tried to establish HSL/p-HSL Westerns, but failed until now. A range of control samples yielded inconsistent and implausible data. Instead, in the revised version of the MS, we now included a transcriptome dataset of the cell models used. From these it is clear, that wildtype and KO cells are virtually indistinguishable in their gene expression patterns. Of >29k genes, only transcripts from 20 are of significantly different abundance, one of which is the UCP1-KO remnant. The other 19 do not include any component of the adrenergic signaling axis or other metabolic components connected to lipolysis (see new supplemental data).

Nevertheless, the observed difference must certainly be caused somewhere and the adrenergic signaling cascade, from receptor to lipases/LD coating proteins, provides a number of promising posttranslational modifications. Systematically probing these to mechanistically clarify the difference in lipolytic rate is a very tempting concept that we will follow up on, but is regrettably beyond the scope of this paper.

5/ The authors derive FAO from O₂ consumption. Why did they rule out that aerobic glycolysis could significantly contribute to O₂ consumption? Most reports to day indicate that BAT is an avid glucose consuming tissue. The contribution of FAO to global O₂ consumption could be estimated using etomoxir, albeit not being a perfect tool.

We did not rule out carbohydrate use. Without the possibility to measure CO₂ production, we do not know the respiratory quotient. Thus, an arbitrary ratio of lipid:carb utilization must be chosen for the model. We chose RQ 0.7 (exclusive lipid ox) for several reasons: it is probably close to reality (definitely closer than 1.0) and it generates “maximal beta-oxidation” values instead of “approximate beta-oxidation”, which seemed easier to interpret. We state this in the Material and Methods section (line 124, “maximal FFA amount”) and use the appropriate wording in the results section (e.g. line 158, “less than 10% of FFA were subject to mitochondrial beta-oxidation”).

Please note that all relevant values delineated from beta-oxidation (e.g. cycling rate and total lipolysis) would only marginally deviate even during massive carb oxidation because the contribution of beta-oxidation to total FFA mobilization is several fold less important than FFA release (which is unaffected by RQ assumptions).

Finally, if beta-oxidation would be overestimated, the case would be even stronger for our central conclusions of cycling rate as an indicator for an alternative thermogenic pathway (because smaller beta-oxidation rates mathematically translate into higher cycling rates).

November 2, 2018

RE: Life Science Alliance Manuscript #LSA-2018-00136-TR

Dr. Tobias Fromme
Technische Universität
Molecular Nutritional Medicine
Gregor Mendel Str. 2
Freising-Weihenstephan 85350
Germany

Dear Dr. Fromme,

Thank you for submitting your revised manuscript entitled "Substrate fluxes in brown adipocytes upon adrenergic stimulation and uncoupling protein 1 ablation". As you will see, the reviewers appreciate the introduced changes, and we would be happy to publish your paper in Life Science Alliance pending final revisions necessary to meet our formatting guidelines:

- please provide individual figure files and move the figure legends into the manuscript text only
- please provide the table as either excel or word file
- please note that there are currently two supplementary S1 figures, please fix
- please add number of replicates and information on error bars for both supplementary figures

A. FINAL FILES:

-- High-resolution figure, supplementary figure and video files uploaded as individual files: See our detailed guidelines for preparing your production-ready images, <http://life-science-alliance.org/authorguide>

B. MANUSCRIPT ORGANIZATION AND FORMATTING:

Full guidelines are available on our Instructions for Authors page, <http://life-science-alliance.org/authorguide>

Sincerely,

Reviewer #1 (Comments to the Authors (Required)):

I think this paper is suitable for publication without further revision.

Reviewer #2 (Comments to the Authors (Required)):

The authors have clarified a number of points and made some valuable conceptual additions based on transcriptomic data. The work is conclusive and the limitations of the study are well balanced in the discussion. There are no further comments from my side.